 

🔓 | **Open Peer Review** | Antimicrobial Chemotherapy | Research Article

# The emergence of resistance to the antiparasitic selamectin in *Mycobacterium smegmatis* is improbable and contingent on cell wall integrity

José Manuel Ezquerra-Aznárez,[1] Henrich Gašparovič,[2] Álvaro Chiner-Oms,[3,4] Ainhoa Lucía,[1] Jesús Blázquez,[5] Iñaki Comas,[4,6] Jana Korduláková,[2] José A. Aínsa,[1,7] Santiago Ramón-García[1,7,8]

**ABSTRACT**  Tuberculosis remains the deadliest infectious disease of the 21st century. New antimicrobials are needed to improve treatment outcomes and enable therapy shortening. Drug repurposing is an alternative to the traditional drug discovery process. The avermectins are a family of macrocyclic lactones with anthelmintic activity active against *Mycobacterium tuberculosis*. However, their mode of action in mycobacteria remains unknown. In this study, we employed traditional mutant isolation approaches using *Mycobacterium smegmatis*, a non-pathogenic *M. tuberculosis* surrogate. We were only able to isolate mutants with decreased susceptibility to selamectin using the Δ*nucS* mutator *M. smegmatis* strain. This phenotype was caused by mutations in *mps1* and *mmpL11*. Two of these mutants were used for a second experiment in which high-level selamectin-resistant mutants were isolated; however, specific mutations driving the phenotypic change to high-level resistance could not be identified. The susceptibility to selamectin in these mutants was restored to the basal level by subinhibitory concentrations of ethambutol. The selection of ethambutol resistance in a high-level selamectin-resistant mutant also resulted in multiple colonies becoming susceptible to selamectin again. These colonies carried mutations in *embB*, suggesting that the integrity of the cell envelope is a prerequisite for selamectin resistance. The absence of increased susceptibility to selamectin in an *embB* deletion strain demonstrated that the target of selamectin is not cytosolic. Our data show that the concurrence of specific multiple mutations and complete integrity of the mycobacterial envelope are necessary for selamectin resistance. Our studies provide first-time insights into the antimycobacterial mode of action of the antiparasitic avermectins.

**IMPORTANCE**  Tuberculosis is the deadliest infectious disease of the 21st century. New antibiotics are needed to improve treatment. However, developing new drugs is costly and lengthy. Drug repurposing is an alternative to the traditional drug discovery process. The avermectins are a family of drugs used to treat parasitic infections that are active against *Mycobacterium tuberculosis*, the bacterium that causes tuberculosis. However, their mode of action in mycobacteria remains unknown. Understanding how avermectins kill mycobacteria can facilitate its development as an anti-mycobacterial drug, including against *M. tuberculosis*.

In this study, we used *Mycobacterium smegmatis*, a non-pathogenic *M. tuberculosis* surrogate model to understand the molecular mechanisms of how selamectin (a drug of the avermectin family selected for this study as a model) acts against mycobacteria. Our data show that the generation of resistance to selamectin is unlikely and that complete integrity of the mycobacterial envelope is necessary for selamectin resistance, providing first-time insights into the antimycobacterial mode of action of the avermectins.

**Peer Reviewers** Mahadi Hassan Mahmoud Abdallah, Alzaiem Alazhari University, Khartoum, Khartoum North, Sudan; Tanjore S. Balganesh, Open Source Drug Discovery, CSIR, Ministry of Science and Technology, Bangalore, Karnataka, India

Address correspondence to Santiago Ramón-García, santiramon@unizar.es.

The authors declare no conflict of interest.

See the funding table on p. 16.

**KEYWORDS** *Mycobacterium smegmatis*, selamectin, repurposing, tuberculosis, mode of action, emergence of resistance

Tuberculosis (TB) is the leading cause of death among infectious diseases during the 21st century. In 2023, an estimated 1.25 million people died from TB, and 10.8 million developed the disease. Despite being curable, the global TB situation has been aggravated in the last decades by the emergence of drug-resistant strains, which currently account for about 3% of newly diagnosed cases and about 20% of the relapses (1). The recent introduction of novel and repurposed anti-TB drugs has allowed the development of shorter and more effective treatments for drug-resistant strains (2–4). However, drug-resistant TB remains the first cause of death due to antimicrobial resistance, and the overall treatment success for drug-resistant strains remains below 60% (1). New anti-TB drugs are thus needed to improve treatment success and prevent the emergence of resistance to the recently introduced compounds.

Developing a new antimicrobial from the bench to the clinic requires enormous time and economic investments, which could take more than a decade and up to more than 2 billion USD (5). Moreover, antimicrobials are often seen as drugs with reduced profitability, discouraging pharmaceutical companies from investing in antimicrobial discovery (6). In recent years, drug repurposing, that is, finding new applications for clinically approved drugs, is gaining momentum as an alternative to the *de novo* discovery process (7). It is estimated that around 4,000 approved drugs are being repurposed for new medical applications (8). Using this approach, it was found that avermectins are active against *Mycobacterium tuberculosis*, the causing agent of TB, and other pathogenic mycobacteria (9–11). This was an unexpected finding given their lack of activity against Gram-positive and Gram-negative bacteria (9).

Avermectins are a family of 16-membered macrocyclic lactones with broad-spectrum anthelmintic activity. They were isolated from *Streptomyces avermitilis* in the 1970s. Ivermectin, a semi-synthetic derivative of the original avermectins, was approved for use in humans in 1987 (12, 13) and it has been used ever since in the prevention of onchocerciasis and lymphatic filariasis (14). Recent studies have identified potential new activities: ivermectin can control glucose and cholesterol levels in diabetic mice, suppress the proliferation of cells in various cancer models, and inhibit *in vitro* the replication of several viruses (15). Furthermore, ivermectin has been proposed as a tool to prevent the transmission of vector-borne diseases, such as malaria (16). Other avermectins are widely used to treat parasite infections in livestock and pets. The antiparasitic mechanism of action of avermectins is well known: avermectins bind to glutamate-gated chloride channels in muscle and nerve cells and increase the permeability to chloride ions; this causes membrane hyperpolarization and eventually leads to death by paralysis (17). Their mode of action against mycobacteria, however, remains unknown.

Selamectin is currently used in veterinary medicine. It displays the most potent *in vitro* activity against mycobacteria and more favorable toxicology profile than other avermectins, making it the most promising candidate to be repurposed as an antimycobacterial drug (9–11). In a previous study, we demonstrated the *in vitro* inhibitory activity of avermectins against the purified *M. tuberculosis* decaprenylphosphoryl-β-D-ribose oxidase (DprE1). However, we did not observe lipid changes associated with DprE1 inhibition in *Mycobacterium smegmatis* and *M. tuberculosis* cells treated with selamectin, and subsequent mutation of key residues for the interaction between DprE1 and avermectins in the *M. smegmatis* chromosome did not affect its susceptibility to selamectin, suggesting that avermectins might interact with multiple targets in addition to DprE1 in mycobacteria (18).

In this study, we report the isolation and characterization of selamectin-resistant mutants and the essential requirements for intermediate and high-level resistance to selamectin in *M. smegmatis*. We leveraged this nonpathogenic, fast-growing mycobacterium as a surrogate for *M. tuberculosis*. While *M. smegmatis* shares the prominent features of the *Mycobacterium* genus, including the mycolic acid-containing cell wall, and

its susceptibility to avermectins is comparable to *M. tuberculosis* (9), the differences in their genome size and niche adaptation make subsequent validation *in M. tuberculosis* necessary, which are beyond the scope of this study.

## RESULTS

### Characterization of selamectin activity against *M. smegmatis*

To establish optimal conditions for mutant isolation and subsequent characterization of selamectin-resistant colonies, we determined the activity of selamectin against the model strain *M. smegmatis* mc²155 in liquid and solid media. The minimal inhibitory concentration (MIC) of selamectin was 4 µg/mL in all media tested, except the one containing the detergent tyloxapol. Because of this, the use of detergents was discarded for assays involving selamectin. Selamectin activity was tested next against different inocula of *M. smegmatis* in a solid 7H10-OADC agar medium. The MIC of selamectin showed a strong inoculum dependence (Table 1).

### Selamectin resistance cannot be selected by traditional mutant isolation approaches in a wild-type genetic background

We first unsuccessfully attempted traditional mutant isolation assays using the wild-type *M. smegmatis* mc²155 strain, with two different outcomes: (i) inoculums below $10^7$ colony-forming unit (CFU) yielded no colonies at selamectin concentrations higher than 4× MIC, whereas (ii) higher inoculums resulted in the growth of bacterial lawns. None of the colonies isolated from the borders of those plates displayed changes in selamectin susceptibility after being propagated in a selamectin-free medium, suggesting that their isolation was due to a transient phenotype. We observed comparable outcomes when the inoculum was a pool of mutants of a transposon library made in *M. smegmatis* HS42 using the TnSPAZ transposon (19) and similar experimental setups. Fifty colonies were tested for susceptibility changes, and none displayed an increased MIC to selamectin. This suggested that no stable genetic mutations associated with selamectin resistance could be selected in a wild-type background. However, internal control mutant isolation assays carried in parallel using 4× MIC isoniazid allowed for the selection of stable resistant mutants with a mutant frequency of $1.8 \cdot 10^{-5}$, which was comparable to previous reports (20).

### Low-level selamectin-resistant mutants can be selected in a mutator genetic background

We subsequently tried to isolate selamectin-resistant mutants using the *M. smegmatis* Δ*nucS* strain, which lacks a functional DNA mismatch repair system and shows mutant frequencies ca. 100-fold higher than wild-type strains (21). We were able to isolate 38 colonies from the mutator strain, while no colonies were recovered in a parallel assay run

**TABLE 1** MIC of selamectin against *M. smegmatis* in different media[a]

| Broth medium | MIC (µg/mL) |
|---|---|
| 7H9-0.2% glycerol-ADC | 4 |
| 7H9-0.2% glycerol-ADC-0.05% tyloxapol | 64 |
| LB | 4 |
| NE | 4 |
| Müller-Hinton II | 4 |
| **Agar medium[b]** | **MIC (µg/mL)** |
| $10^3$ | 4 |
| $10^5$ | 8 |
| $10^7$ | 16–32 |

[a]MICs were determined in three independent experiments using technical duplicates. the most frequent value is depicted in the table.
[b]MIC in solid medium was tested on 7H10-OADC agar. The activity of selamectin was tested against different starting inocula ($10^3$, $10^5$, $10^7$ CFU).

with the reference *M. smegmatis* mc$^2$155. The mutant frequency was $1.0\cdot10^{-6}$, whereas isoniazid displayed a mutant frequency of $5.2\cdot10^{-4}$ at 4× MIC (Table S1).

We subsequently determined the MIC of the 38 colonies and found that two of them displayed a twofold increase and 23 had a 1.4-fold increase in their MIC to selamectin. Despite being modest variations, results were consistent across several independent assays. Thus, we performed additional validation studies using drop-dilution (Fig. 1A) and time-kill kinetics (Fig. 1B) assays. Both assays correlated well: clones displaying increased survival on selamectin-containing agar plates were also less susceptible to selamectin on time-kill kinetic assays, thus confirming the subtle susceptibility changes observed.

## Concurrence of nonsense mutations in *mps1* and *mmpL11* is necessary for intermediate selamectin resistance

We then performed whole-genome sequencing (WGS) of 10 selected clones (referred to as SEL-I1 to SEL-I10), which revealed the presence of a nonsense mutation in *mps1* (*MSMEI_0391*), encoding a polyketide synthase required for the biosynthesis of glycopeptidolipids (GPLs), in 5 out of the 10 clones. One of them harbored an additional nonsense mutation in the *mmpL11* (*MSMEI_0234*) locus, which encodes a membrane lipid transporter. Additional non-synonymous mutations are shown in Table S2.

Given the sheer number of mutations identified, genetic validation efforts were focused on the two nonsense mutations. Single-stranded DNA recombineering was used to introduce them in the parental *M. smegmatis* Δ*nucS* strain. The reason to use this strain was the bias of NucS toward the correction of G-T, T-T, and G-G mismatches (22), which were predominant among the SNPs found in the sequenced mutants. The first recombineering attempt to reproduce the two nonsense mutations was carried out using

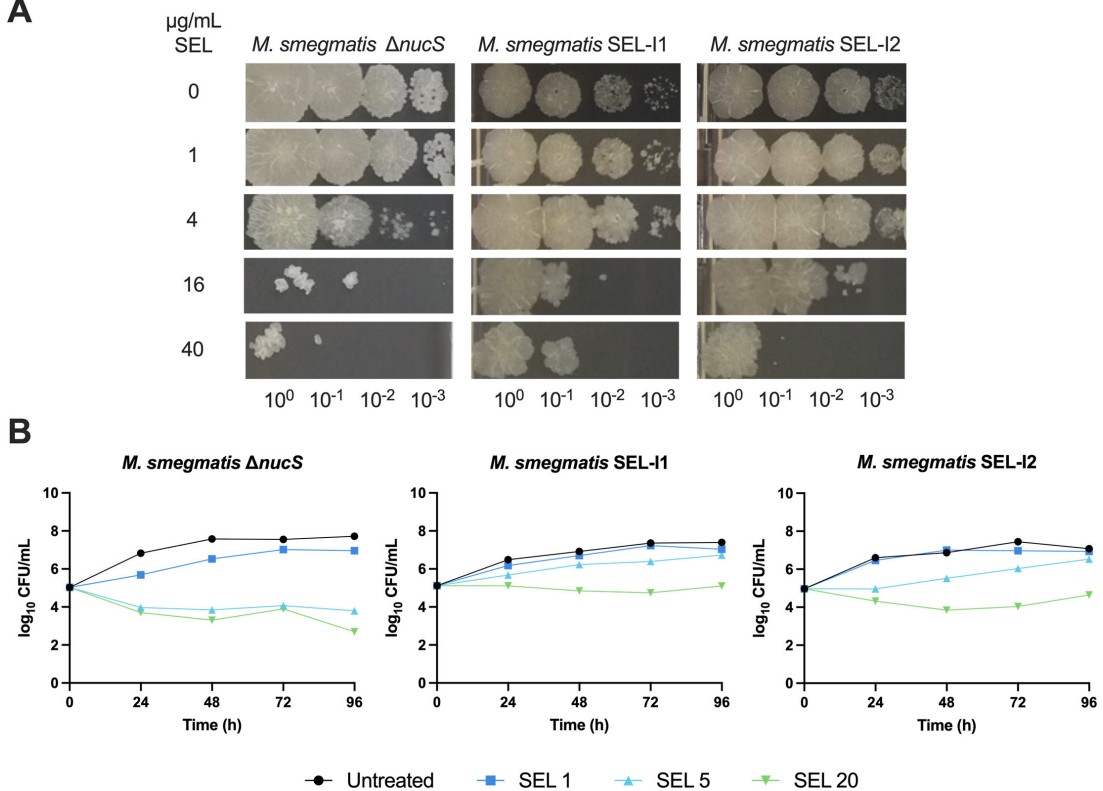

**FIG 1** Selamectin resistance in mutants isolated from the mutator strain *M. smegmatis* Δ*nucS*. Both SEL-I1 and SEL-I2 carry nonsense mutations in *mps1*, and SEL-I1 carries an additional nonsense mutation in *mmpL11*. (A) Drop dilution assay 10-fold serial dilutions of the bacterial inoculum were made from left to right. The figure shows a representative result of two biological replicates. (B) Time-kill kinetics assay. The figure shows a representative replicate with CFUs enumerated in technical duplicates. Selamectin (SEL) concentrations are expressed in µg/mL.

selamectin as the selectable marker. However, no colonies with the desired mutations were retrieved when using ssDNA substrates encoding the nonsense mutations. This observation aligned with the results from mutant isolation assays carried out with the wild-type *M. smegmatis* mc[2]155, in which plating large inoculums resulted in the isolation of colonies without stable resistance phenotypes to selamectin.

Thus, selamectin selection was replaced by streptomycin (Sm) resistance as co-selection. This approach allowed the introduction of the *mps1* and *mmpL11* nonsense mutations in the *M. smegmatis* Δ*nucS* background. Whereas only the *mmpL11*-engineered strain displayed a subtle increase in the MIC of selamectin in broth, both mutants displayed slightly decreased susceptibility to selamectin in drop dilution assays (Fig. 2; Table 2).

A double mutant carrying both nonsense mutations was built to assess whether these mutations had an additive effect on selamectin resistance. The coexistence of both mutations reproduced the phenotype observed in the intermediate SEL-I1 mutant, and the double mutant was less susceptible compared to each of the single mutants (Fig. 2).

## High-level resistant mutants to selamectin require mutations in different loci

Following the genetic validation of *mps1* and *mmpL11* mutations, a new mutant isolation assay was set using two of the previously isolated mutants: SEL-I1 (carrying nonsense mutations in *mps1* and *mmpL11*, among other point mutations) and SEL-I2 (carrying the nonsense mutation in *mps1*, among other point mutations). The MIC of selamectin was tested in 10 colonies derived from each mutant, displaying MIC >32 µg/mL. This MIC shift was confirmed by time-kill kinetics in 9 out of the 10 total mutants tested (Fig. 3; Fig. S1).

WGS identified differences between the two genetic backgrounds. On the one hand, the four clones isolated from SEL-I1 (named SEL-R1 to SEL-R4) carried identical mutations, suggesting they derived from a single clone (Table S3), with two additional nonsense mutations in *mshA* (encoding a D-inositol-3-phosphate glycosyltransferase enzyme in the mycothiol biosynthetic pathway) and *MSMEI_1249* (conserved hypothetical protein). On the other hand, the five clones isolated from SEL-I2 (named SEL-R5 to SEL-R9) carried different mutations from each other (Tables S4 and S5). Four of them shared a missense mutation in *mmpL11*, which confirmed the role of this locus in the acquisition of selamectin resistance and that *mps1* mutations were likely the first step in the process of acquiring resistance. Another mutation shared by several clones isolated from SEL-I2 was the substitution of the canonical start codon to the less efficient GUG in *mspA*, the gene encoding for the major mycobacterial porin in *M. smegmatis*.

Mutations in *mshA* and *MSMEI_1249* were introduced separately in the *M. smegmatis* Δ*nucS* background. Neither of them reproduced the high-level selamectin resistance (Fig. S2). Then, both mutations were introduced sequentially in the engineered strain carrying *mps1* and *mmpL11* nonsense mutations. Similarly, they had no effect on the susceptibility to selamectin, as the increased survival observed at 4 µg/mL was due to

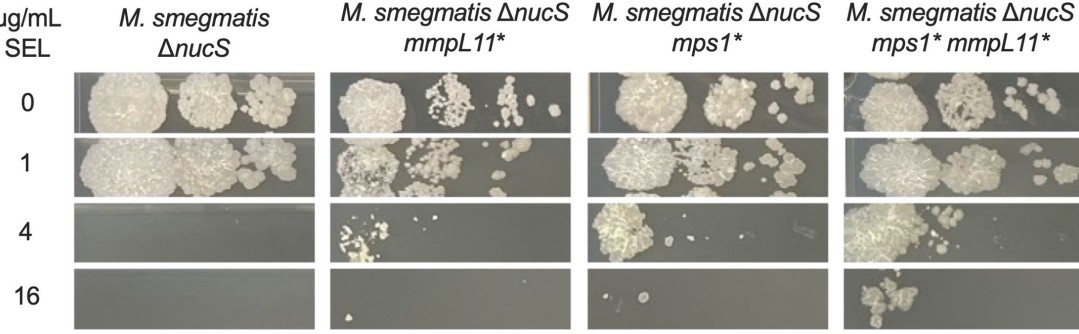

**FIG 2** Phenotypic validation of *M. smegmatis* engineered point mutants at *mps1* and *mmpL11* by drop dilution assays. Selamectin (SEL) concentrations are expressed in µg/mL. 10-fold serial dilutions of the bacterial inoculum were made from left to right. *Truncated protein due to a stop codon in the ORF. The figure shows a representative result of three biological replicates.

**TABLE 2** MIC of selamectin for *M. smegmatis* Δ*nucS* engineered strains[a]

| Strain | MIC (µg/mL) |
|---|---|
| *M. smegmatis* Δ*nucS* | 4 |
| *M. smegmatis* Δ*nucS mps1** | 4 |
| *M. smegmatis* Δ*nucS mmpL11** | 5.6 |
| *M. smegmatis* Δ*nucS mps1*, mmpL11** | 5.6 |
| *M. smegmatis* Δ*nucS mshA** | 5.6 |
| *M. smegmatis* Δ*nucS MSMEI_1249** | 5.6 |
| *M. smegmatis* Δ*nucS mps1*, mmpL11*, MSMEI_1249*, mshA** | 5.6 |

[a]MICs were determined in two independent experiments using technical duplicates. the most frequent value is depicted. *The protein product of the corresponding gene was truncated by the introduction of a STOP codon.

the previously introduced mutations responsible for the SEL-I phenotype (Fig. S2; Table 2).

To study the role of porins in the susceptibility of *M. smegmatis* to selamectin (identified in SEL-I2, *mspA*), the MIC of selamectin was determined against porin-deficient *M. smegmatis* strains. We speculated that the substitution identified at the start codon would reduce the amount of protein; however, porin-deficient mutants remained susceptible to selamectin as the parental strain (Table S6).

## Selamectin-resistant mutants display altered composition of glycopeptidolipids

WGS of selected intermediate resistant strains (SEL-I1 and SEL-I2) and derived high-level resistant mutants (R1-R9) revealed the presence of nonsense mutations in genes *mps1* and *mmpL11*, both involved in the biogenesis and integrity maintenance of the mycobacterial cell envelope. In *M. smegmatis*, *mps1* encodes a non-ribosomal peptide synthase playing a role in the biosynthesis of GPLs, and *mmpL11* is described as a part of lipid export machinery (23–27). To confirm whether the aforementioned mutations induce alterations in the cell envelope of tested *M. smegmatis* strains, total lipids, alkaline-stable GPLs, and fatty/mycolic acids were extracted and analyzed from wild-type and mutant strains grown in 7H9 broth at 37°C. Compared to wild type, thin-layer chromatography (TLC) analyses did not reveal significant changes in the composition of mycolic and fatty acids, mycolic acids-derived molecules, phosphatidylethanolamine, cardiolipin, triacylglycerols or phosphatidylinositol mannosides in selamectin-resistant strains. However, in correlation with results obtained by WGS, resistant strains carrying mutations in gene *mps1* or in genes *mps1/mmpL11* displayed notable alterations in the production of GPLs (Fig. 4E).

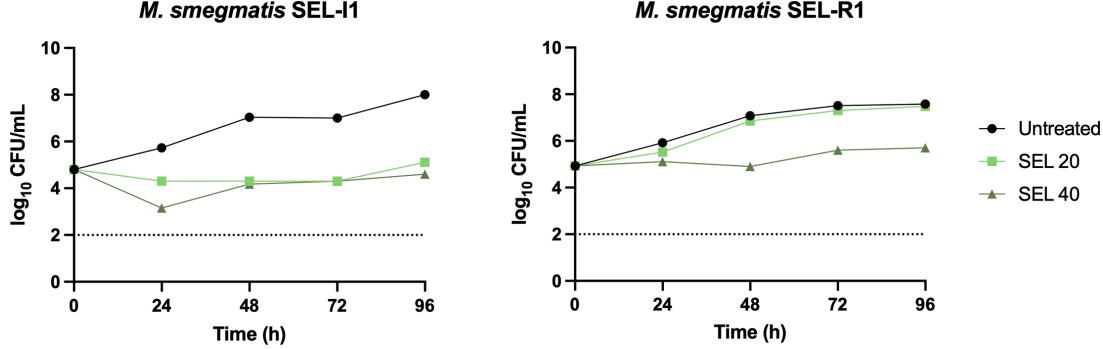

**FIG 3** Time-kill kinetics of selamectin against SEL-R1, a representative high-level selamectin-resistant clone, and its parental strain (SEL-I1). Selamectin (SEL) concentrations are given in µg/mL. The figure shows a representative replicate. CFUs were enumerated in technical duplicates.

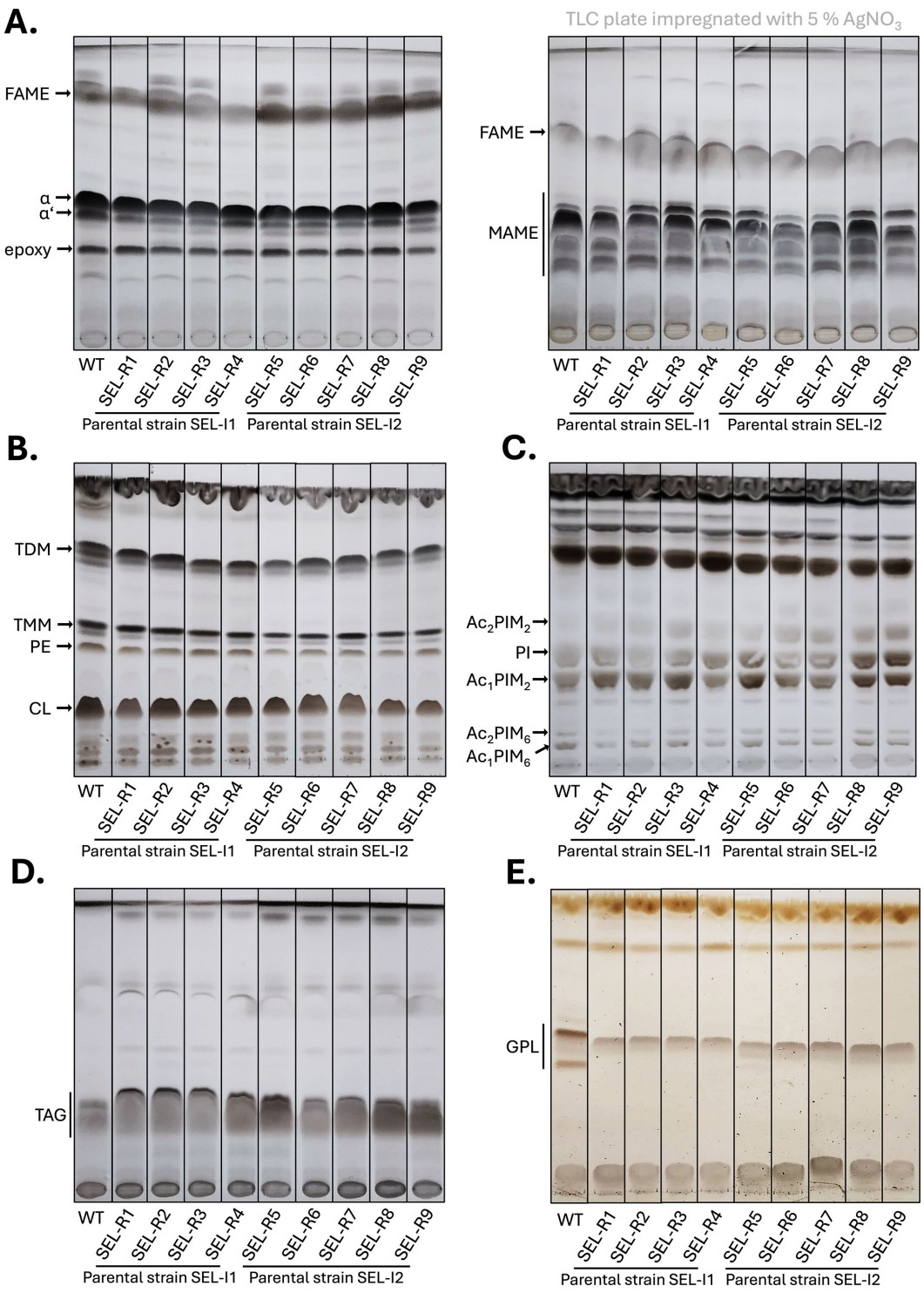

**FIG 4** TLC analysis of mycolic/fatty acids and lipids isolated from selamectin-resistant strains of *M. smegmatis*. (A) TLC analyses of mycolic and fatty acids methyl esters isolated from whole cells. Samples were loaded on standard (left) or AgNO$_3$-impregnated (right) TLC plates and separated in solvent V. Isolated lipids were separated in (B) solvent I, (C) solvent II and (D) solvent III. (E) Glycopeptidolipids were separated in solvent IV. Lipids and mycolic/fatty acids were visualized with 10% CuSO$_4$ in 8% phosphoric acid solution and charring. Glycopeptidolipids were visualized with orcinol in H$_2$SO$_4$ followed by charring. FAME, fatty acids methyl esters; MAME, mycolic acids methyl esters; ɑ, ɑ' and epoxy refer to forms of mycolic acids methyl esters; TDM, trehalose dimycolates; TMM, trehalose monomycolates; PE, phosphatidylethanolamine; CL, cardiolipin; Ac$_2$PIM$_2$, diacylated phosphatidylinositol dimannosides; PI, phosphatidylinositol; Ac$_1$PIM$_2$, monoacylated phosphatidylinositol dimannosides; Ac$_2$PIM$_6$, diacylated phosphatidylinositol hexamannosides; Ac$_1$PIM$_6$, monoacylated phosphatidylinositol hexamannosides; TAG, triacylglycerols; GPL, glycopeptidolipids.

## Subinhibitory concentrations of cell wall inhibitors restored susceptibility to selamectin in the SEL-R1 background

Intermediate selamectin resistance appears to be driven by mutations that affect cell wall composition, and those changes seem necessary to develop high-level resistance. We then hypothesized that the combination of selamectin with cell wall targeting compounds could restore the susceptibility to this drug in the mutant background. To validate this hypothesis, the MIC of selamectin against the mutant SEL-R1 and the wild-type strain *M. smegmatis* mc$^2$155 was determined in the presence of subinhibitory concentrations (0.25× MIC) of compounds targeting peptidoglycan (vancomycin, D-cycloserine, imipenem, cefradine, amoxicillin), arabinogalactan (ethambutol, BTZ043), and mycolic acid (isoniazid) biosynthesis. Compounds could be divided into three groups based on their effect on the MIC of selamectin in both strains (Table 3). First, most compounds showed no interaction with selamectin (i.e., the MIC of selamectin did not change in the presence of subinhibitory concentrations of these compounds). The second group (rifampicin, kanamycin) produced similar fold changes in the MIC in both strains, suggesting that their interaction with selamectin was independent of the mutations in the SEL-R1 strain. A third group, represented by ethambutol, vancomycin, and cefradine, showed a differential interaction with the SEL-R1 background, which resulted in the restoration of the susceptibility to selamectin.

We then studied in more detail the interaction between selamectin and the first-line anti-TB drug ethambutol. For the reference *M. smegmatis* mc$^2$155 strain, the MIC of selamectin did not change in the presence of ethambutol, and no interaction was detected using the checkerboard assay, in which a Fractional Inhibitory Concentration Index (FICI) of 0.75 was obtained. By contrast, in the SEL-R1 mutant background, the interaction between selamectin and ethambutol was confirmed with a FICI of 0.25, indicative of synergism (Fig. 5).

Ethambutol was thus able to restore selamectin wild-type susceptibility levels in a selamectin-resistant mutant. We speculated whether the opposite effect (i.e., selamectin reducing the MIC of ethambutol in an ethambutol-resistant mutant) could also be possible. For this, ethambutol-resistant mutants were generated from two different backgrounds: (i) the *M. smegmatis* Δ*nucS* (selamectin-susceptible) and (ii) the SEL-R1 selamectin-resistant mutant (a derivative of *M. smegmatis* Δ*nucS*). Sanger sequencing revealed that all 14 ethambutol-resistant mutants selected—six from *M. smegmatis* Δ*nucS* and eight from *M. smegmatis* SEL-R1—carried mutations in *embB*, encoding one of the three arabinosyltransferases targeted by ethambutol, although at different positions (Table 4). We then tested the activity of selamectin against the newly generated *embB* mutants. Susceptibility to selamectin in the mutants derived from *M. smegmatis* Δ*nucS* did not change compared to the parental strain. However, the acquisition of ethambutol resistance in the SEL-R1 background resulted in decreased MIC values of selamectin in six out of the eight colonies tested (Table 4). Time-kill kinetics of four strains representing all possible combinations of susceptibility to selamectin and ethambutol showed that subinhibitory concentrations of ethambutol could make selamectin subMIC concentrations become bactericidal, as it was the case for SEL-R1 and EMB-R5 (Fig. 6). Subinhibitory concentrations of selamectin, however, did not increase the activity of ethambutol in EMB-R11, which behaved as the parental strain. The role of EmbB in *M. smegmatis* susceptibility to selamectin was assessed by constructing an *embB* knockout strain in the mc$^2$155 background and an *embB* overexpression strain. However, neither of them affected significantly the susceptibility to selamectin (Table S7).

## DISCUSSION

In recent years, drug repurposing has emerged as a cost-effective alternative to the *de novo* drug discovery process. Following this strategy, a phenotypic screening identified the activity of the avermectins against *M. tuberculosis* and other mycobacteria (9–11). Among avermectins, selamectin showed the best potential against mycobacteria. Selamectin is a veterinary avermectin used in cats and dogs, with improved

**TABLE 3** Variations in the MIC of selamectin in the presence of 0.25× MIC of cell wall inhibitors[a]

| Compound | Fold change (*M. smegmatis* mc$^2$155) | Fold change (*M. smegmatis* SEL-R1) |
|---|---|---|
| Cefradine | 2 | 16 |
| Ethambutol | 1 | 8 |
| Vancomycin | 4 | 8 |
| D-cycloserine | 1 | 1 |
| Imipenem | 1 | 1 |
| Isoniazid | 1 | 1 |
| Amoxicillin | 1 | 1 |
| BTZ043 | 1 | 1 |
| Rifampicin | 4 | 4 |
| Kanamycin | 2 | 2 |

[a]Fold change is calculated as the ratio between the MIC of SEL alone divided by the MIC of SEL in the presence of 0.25× MIC of a given compound. The MIC of selamectin was 2 µg/mL for *M. smegmatis* mc$^2$155, and 8–16 µg/mL for *M. smegmatis* SEL-R1. Kanamycin and rifampicin were included as controls because they target processes that are not directly related to the cell wall. MICs were determined in two independent experiments using technical duplicates. The most frequent value is depicted in the table.

pharmacokinetic properties and lower toxicity over other avermectins (28). Understanding its mechanism of action can contribute to accelerate the progress through the drug discovery pipeline. In this study, we used mutant isolation approaches to study the mechanism of action of the avermectins against *M. smegmatis*, a non-pathogenic surrogate of *M. tuberculosis*.

Mutant isolation attempts with the wild-type *M. smegmatis,* and a transposon mutant pool allowed the isolation of colonies on high selamectin concentrations, but none of them displayed a stable resistant phenotype. This limitation was overcome using the mutator *M. smegmatis* Δ*nucS* strain, which allowed the isolation of intermediate selamectin-resistant mutants (SEL-I with modest increases in their MIC) (Fig. 1). WGS of SEL-I clones revealed the presence of nonsense mutations in two genes (*mmpL11* and *mps1*), both of which are involved in the biogenesis of the *M. smegmatis* envelope. GPLs are polar lipids exclusively found on the outer membrane of rapid-growing mycobacteria. Their presence or absence determines the overall hydrophobicity of the mycobacterial envelope and therefore affects properties such as the ability to form biofilms. The lack of GPLs has been associated with aggregation and cording in fast-growing mycobacteria (29), which is consistent with the phenotype observed for the SEL-I mutants with nonsense mutations in the *mps1* gene. Analysis of the lipid profile of SEL-R mutants carrying the nonsense mutations in *mps1* confirmed that they were devoid of GPLs (Fig. 4). Deletion of *mmpL11* results in reduced membrane permeability, resistance to antimicrobial peptides, and alterations in biofilm formation. Its role seems to be conserved in *M. tuberculosis*, where it is required for full virulence in mice (23–25); however, caution needs to be taken to extrapolate this observation to *M. tuberculosis*, which lacks GPLs. The introduction of both mutations in *M. smegmatis* Δ*nucS* reproduced the observed phenotype of the SEL-I strains (Fig. 2), confirming their contribution to the intermediate resistance phenotype. We hypothesize that the changes in the composition of the outer membrane could be affecting the permeability across the mycobacterial envelope, thus affecting selamectin susceptibility. The increased clumping tendency resulting from these mutations could promote the reduction of selamectin susceptibility, if bacteria growing within the clumps were exposed to lower selamectin concentrations (Fig. S3).

The SEL-I strains provided a genetic background to isolate high-level selamectin-resistant mutants. WGS of SEL-R mutants provided a possible path for the development of the SEL-I phenotype: that is, mutants derived from SEL-I2 acquired a missense mutation in *mmpL11*. It was not possible, however, to identify specific single mutations conferring high-level resistance. This was a limitation of the use of a mutator strain, where mutations conferring no adaptive advantages are more easily co-selected.

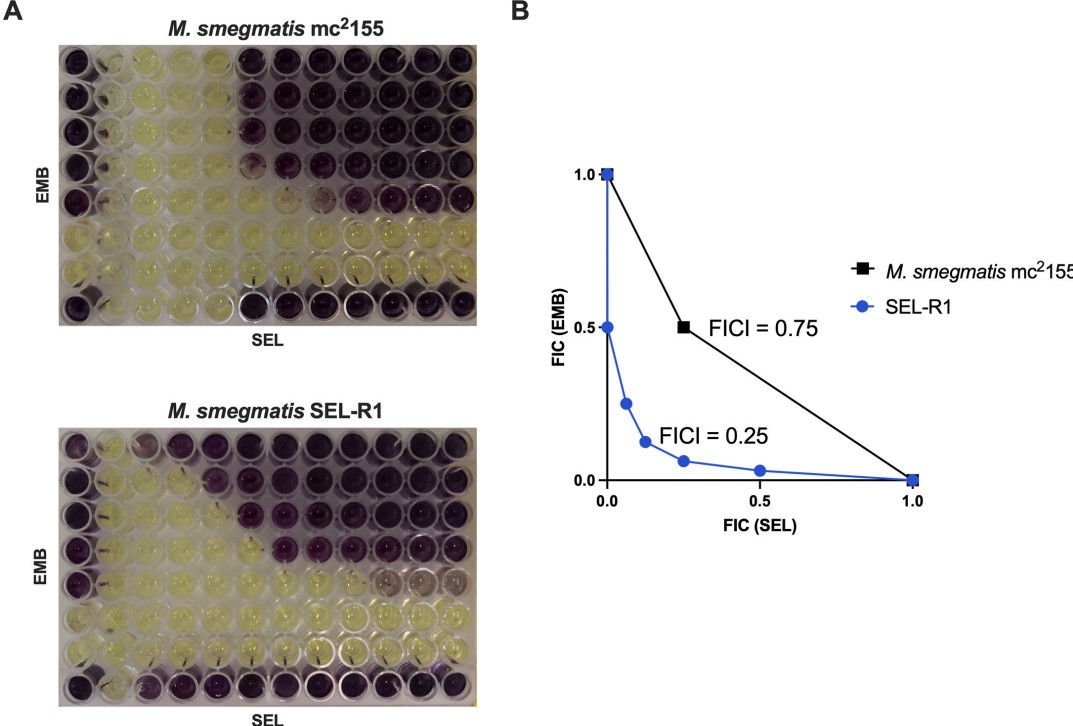

**FIG 5** Selamectin and ethambutol interaction studies against *M. smegmatis*. (A) Wild-type *M. smegmatis* mc²155 and mutant *M. smegmatis* SEL-R1 strains were tested by the checkerboard assay. Dark violet wells indicate bacterial growth, while yellow wells indicate growth inhibition. The maximum concentrations used were 4 µg/mL for ethambutol and 32 µg/mL for selamectin. (B) Isobologram plot for the selamectin-ethambutol interaction in each genetic background. The figure shows a representative result of two independent experiments with technical duplicates.

Selamectin resistance was reversed in the presence of subinhibitory concentrations of the cell wall inhibitors ethambutol, vancomycin, and cefradine (Table 3). Ethambutol targets the arabinosyltransferases EmbA and EmbB, involved in the synthesis of the arabinogalactan chains; and EmbC, involved in lipoarabinomannan biosynthesis (30, 31). Vancomycin and cefradine prevent peptidoglycan crosslinking by binding to the peptidoglycan pentapeptides, or by inactivating penicillin-binding proteins, respectively. The selamectin-ethambutol interaction was then studied in more depth using mutants with different susceptibilities to both drugs. Selection for ethambutol resistance using the SEL-R1 background yielded colonies that had reversed selamectin resistance. The EmbB Q483R mutation was the only mutation that did not restore selamectin susceptibility in SEL-R1-derived mutants. This mutation was the only one associated with clinical ethambutol resistance in *M. tuberculosis*, suggesting that mutations that restore selamectin susceptibility might have significant fitness costs on *M. smegmatis*. The fact that selamectin susceptibility can be restored by chemical or genetic inhibition of the mycobacterial envelope assembly suggests that mutations involved in the acquisition of high-level resistance are more likely impeding the access of selamectin to its target(s) rather than affecting the interaction between them and selamectin.

There are several cases in which selection of mutations on the target is not possible or does not lead to resistance, that can be extrapolated to selamectin. First, selamectin could be acting on multiple targets. In this case, mutations in only one target would not confer resistance, owing that other targets would still be susceptible. This would be consistent with our previous findings in which mutations in DprE1 (that altered the affinity of selamectin toward the enzyme) did not affect *M. smegmatis* susceptibility to selamectin (18). Second, selamectin could be inhibiting multiple targets carrying out the same process. In this case, it is possible that the susceptible target dominates over the resistant one, as it occurs for aminoglycoside resistance in *Escherichia coli* expressing

**TABLE 4** Characterization of ethambutol-resistant mutants from the *M. smegmatis* SEL-R1 and Δ*nucS* strain backgrounds[a]

| | | | MIC (µg/mL) | |
|---|---|---|---|---|
| Background | Mutant name | EmbB mutation | SEL | EMB |
| *M. smegmatis* SEL-R1 | SEL-R1* | wild type | 16 | 2 |
| | EMB-R2 | Y320H | 4 | 16 |
| | EMB-R3 | G310R | 1 | 8 |
| | EMB-R4 | Y320H | 8 | 16 |
| | EMB-R5 | Q483R | 16 | 16 |
| | EMB-R6 | Q483R | 16 | 16 |
| | EMB-R7 | L388P | 2 | 16 |
| | EMB-R8 | G310R | 1 | 4 |
| | EMB-R9 | L388P | 2 | 16 |
| *M. smegmatis* Δ*nucS* | Δ*nucS** | wild type | 2 | 2 |
| | EMB-R11, 13, 15, 16, 17, 18 | R307H | 2 | 16 |

[a]Parental strains used for mutant isolation are marked with (*). Both parental strains were sequenced and showed no mutations when aligned to the reference *embB* sequence. EMB, ethambutol. SEL, selamectin. MICs were determined in two independent experiments using technical duplicates. The most frequent MIC value is depicted in the table.

different rRNA operon alleles (32), or bedaquiline susceptibility in strains expressing bedaquiline-susceptible and resistant *atpE* alleles simultaneously (33). Third, the target might not be a protein and therefore its modification to become insensitive to selamectin requires the modification of a complete metabolic route, as it occurs with vancomycin, against which resistance can be achieved in *Staphylococcus aureus* by modifying the

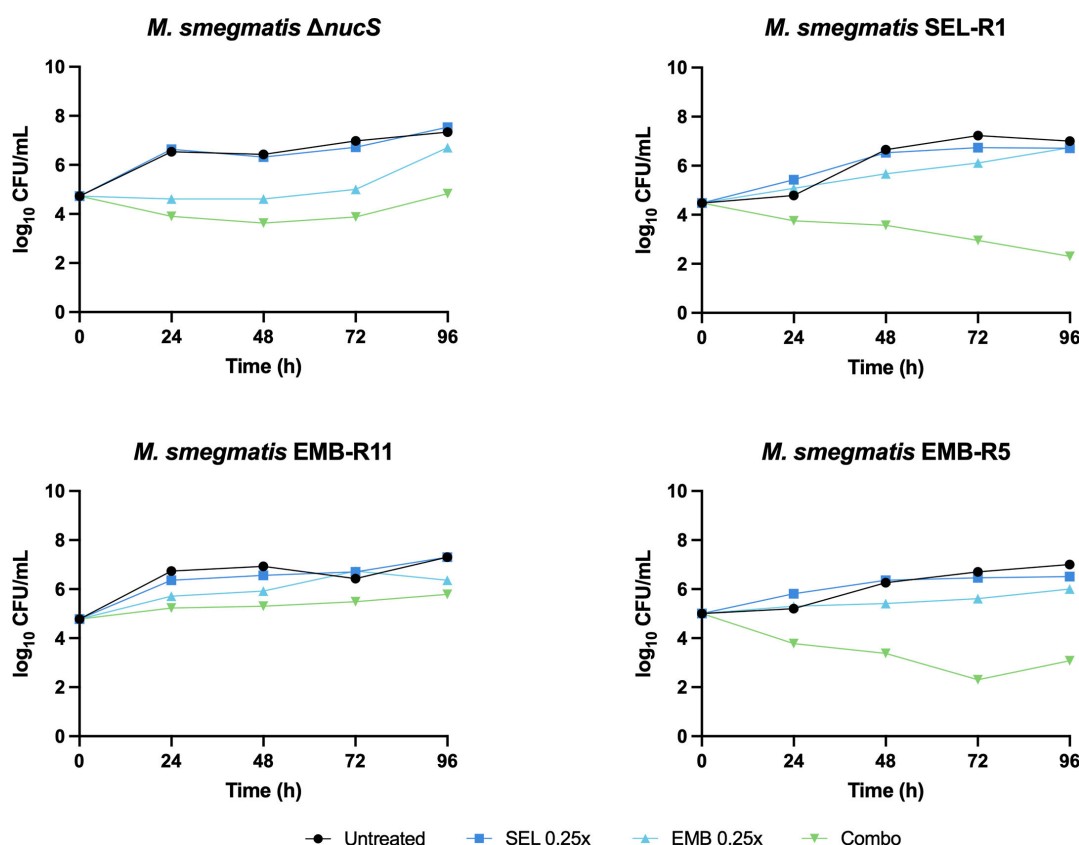

**FIG 6** Time-kill kinetics of *M. smegmatis* mutants in the presence of combinations of selamectin and ethambutol at subinhibitory concentrations. Compounds were tested at 0.25-fold the MIC of the corresponding strain. The figure shows a representative replicate. CFUs were enumerated in technical replicates.

D-Ala-D-Ala moiety from peptidoglycan precursors but requires alternative enzymes that recognize the modified precursors (34).

Altogether, our results from mutant isolation assays and subsequent genetic and phenotypic characterization prove that the generation of high-level selamectin resistance is highly unlikely and requires multiple step mutations to occur, which can be the result of selamectin acting on multiple targets. Being selamectin refractory to the generation of resistant mutants, this feature could be of great value in the development of novel regimens for TB treatment. Further studies are needed to understand the mode of action of selamectin against the pathogenic *M. tuberculosis* and how to best use it in combinatorial regimens.

## MATERIALS AND METHODS

### Bacterial strains, compounds, and culture conditions

*M. smegmatis* strains were routinely grown at 37°C in Middlebrook 7H9 broth (Difco) supplemented with 10% Middlebrook Albumin-Dextrose-Catalase (ADC) (Difco) and 0.05% (vol/vol) Tyloxapol or on Middlebrook 7H10 agar (Difco) supplemented with 10% (vol/vol) Middlebrook Oleic Acid-Albumin-Dextrose-Catalase (OADC) (Difco). Plasmids used in this study are listed in Table S8.

For strains carrying selectable markers, antibiotics were added at the following concentrations: kanamycin (Km, Sigma), 20 µg/mL; hygromycin B (Hyg, Invivogen); 20 µg/mL, streptomycin (Sm, Sigma), and 20 µg/mL. Selamectin was purchased from the European Pharmacopoeia.

### Drug susceptibility assays

MICs were determined in 7H9 broth supplemented with 0.2% glycerol and 10% ADC, using twofold serial dilutions of the compounds in duplicates in 96-well flat-bottomed polystyrene plates. Plates were inoculated with $10^5$ CFU/mL of *M. smegmatis* and incubated in the presence of the compounds for 3 days before the addition of 30 µL of a solution of the bacterial growth reporter MTT [3-(4,5-dimethylthiazol-2-yl)−2,5-diphe-nyltetrazolium bromide] (2.5 mg/mL) and Tween 80 (10%, vol/vol). Following 3 hours of incubation, the optical density at 580 nm ($OD_{580}$) was measured, and the MIC was defined as the lowest concentration of compound that inhibited at least 90% of MTT conversion to formazan.

*M. smegmatis* MICs were also determined on 7H10 agar with 10% OADC, testing twofold serial dilutions of the compounds in duplicates in 24-well polystyrene plates. Compounds were added to tempered agar and mixed thoroughly in 15 mL centrifuge tubes before transferring 2 mL to each well. Three different inoculums, $10^3$, $10^5$, and $10^7$ total CFU, were seeded onto each well by transferring 10 µL of bacterial suspension and incubated for 3 days. The MIC was defined as the minimal concentration of compound which inhibited colony formation.

### Checkerboard assays

Checkerboard assays were performed as previously described (35). Compound stocks were prepared at 4× the highest test concentration for each compound. Then, the two compounds were diluted by making twofold dilutions across all rows and columns, respectively. Plates were inoculated with $10^5$ CFU/mL, and the readout was performed as described for MIC determinations. The fractional inhibitory concentrations (FIC) were defined as the ratio between the MIC of either compound in the combination and the MIC of the compound alone. Then, FIC indices (FICI) were obtained by adding each drug FIC, and the lowest FICI value was used to determine whether an interaction was synergistic (FICI ≤0.5), there was no interaction (0.5 < FICI < 4) or was antagonistic (FICI ≥4) (36).

## Time-kill kinetic assays

Culture flasks of 25 cm$^2$ containing 10 mL of 7H9 with 0.2% glycerol (vol/vol) and 10% ADC were inoculated with *M. smegmatis* to a final density of 10$^5$ CFU/mL from liquid cultures in the late exponential phase. Aliquots of the culture were taken at 0, 6, 24, 48, 72, and 96 hours following the addition of the compounds; at each timepoint, 10-fold serial dilutions were made in PBS with 0.1% tyloxapol and 100 µL was seeded onto Luria-Bertani (LB) agar plates. CFUs were determined after 3 days of incubation at 37°C.

## Selection of selamectin-resistant mutants

Mutant isolation was first attempted by plating 10$^5$ to 10$^8$ total CFUs of *M. smegmatis* mc$^2$155 or a transposon library made in *M. smegmatis* HS42 using the TnSPAZ transposon (19) onto 90 mm Petri dishes containing 1-, 2-, 4-, 10-, or 20-fold the MIC of the selamectin against the corresponding inoculum adjusted per surface unit. Because the surface of a 90 mm Petri dish is roughly 100 times the surface of a 10 µL droplet, the MIC value chosen as reference for each inoculum was the one corresponding to a 100-fold lower inoculum in a 24-well plate.

Mutant isolation attempts with the *M. smegmatis* ΔnucS strain (21) were performed under conditions that restricted *M. smegmatis* mc$^2$155 growth (4- to 20-fold the MIC and inoculums below 10$^7$ CFUs). Two of these colonies (SEL-I1 and SEL-I2) were subsequently used for the isolation of high-level resistance on plates containing 10 or 20-fold the MIC of selamectin.

In all cases, plates were incubated at 37°C for up to 7 days. For their phenotypic validation, the colonies isolated were transferred to a drug-free medium and their susceptibility to selamectin was determined using three different assays. First, the MIC of selamectin against each strain was determined as above described but using 1.4-fold dilution steps instead of conventional two-fold dilution steps. Second, susceptibility to selamectin was tested on agar plates by seeding 5 µL spots of four 10-fold serial dilutions (starting from 10$^7$ bacteria/mL, i.e., 5·10$^4$ total bacteria) onto plates containing 0, 1, 4, 16, or 40 µg/mL of selamectin. In a final validation step, resistance to selamectin was confirmed by time-kill kinetic assays. Those clones with decreased selamectin susceptibility were selected for WGS.

## Selection of ethambutol-resistant mutants

Ethambutol-resistant mutants were selected by plating approximately 10$^7$ CFUs of *M. smegmatis* onto 90 mm 7H10-OADC plates containing 4-fold the MIC of ethambutol on solid media. The colonies isolated were then transferred to a drug-free medium and their resistant phenotype was confirmed by MIC determination in a liquid medium.

## Whole-genome sequencing and identification of polymorphisms

Genomic DNA from the resistant *M. smegmatis* mutants was extracted using the CTAB (Cetyl Trimethyl Ammonium Bromide) method (37). Briefly, bacteria were resuspended in 400 µL TE (100 mM Tris/HCl, 10 mM EDTA, pH = 8.0) and heated at 85°C for 10 min. Samples were then chilled and treated with 50 µL of lysozyme (10 mg/mL) at 37°C for at least 1 hour. Hereafter, 72.5 µL of 10% sodium dodecyl sulfate and 2.5 µL of proteinase K (20 mg/mL) were added and the resulting mixture was incubated at 65°C for 10 min. Subsequently, 100 µL of 5 M NaCl and 100 µL of prewarmed CTAB/NaCl (10% CTAB in 0.7 M NaCl) were added and incubated at 65°C for a further 10 min. Genomic DNA was extracted by adding 750 µL of chloroform:isoamyl alcohol (24:1). Samples were centrifugated (5 min, 12,000 RCF) and the upper phase was transferred to a fresh tube containing 450 µL isopropanol. Nucleic acids were precipitated overnight at −20°C and were collected by centrifugation (10 min, 12,000 RCF). The pellets were dissolved in 50 µL nuclease-free water and quantified using an ND-1000 spectrophotometer.

Genomic DNA was sequenced at the FISABIO Sequencing and Bioinformatics Service (Valencia, Spain). DNA libraries were prepared using the Nextera XT DNA Library Prep

kit (Illumina) following the manufacturer's instructions and sequenced on the Illumina MiSeq platform. The Illumina paired-end reads were filtered with fastp (38) to remove low-quality bases at the 3′ ends. The filtered reads were subsequently mapped to the *M. smegmatis* chromosome (GenBank accession number CP001663.1) with BWA (39) and potential duplicates were removed with Picard tools (http://broadinstitute.github.io/picard). Single nucleotide polymorphisms (SNPs) were called with VarScan (40) if at least 20 reads supported the genomic position, the SNP was found at a frequency of 0.9 or higher and was not found near an indel region (10 bp). Indels were called with Genome Analysis ToolKit (GATK) (41). SNPs and indels were annotated by using SnpEFF (42), and those common to the parental strain were later removed.

## Genetic manipulation of *M. smegmatis*

*M. smegmatis* single-point mutants were constructed using single-stranded DNA recombineering (43). Mutagenic oligonucleotides were designed so that they would introduce additional silent mutations to generate a new restriction site (Table S9).

Recombineering-induced *M. smegmatis* were prepared from cultures grown to the mid-exponential phase ($OD_{600}$ = 0.5). At that point, acetamide was added to a final concentration of 0.2% (wt/vol), and the culture was incubated for an additional 4 hours to allow induction of the recombineering system. Bacteria were then transferred to the ice for 30 min, harvested (5 min, 4,000 RCF), washed four times with ice-cold 10% glycerol, and resuspended in 1/100 of the initial culture volume.

Recombineering-induced *M. smegmatis* Δ*nucS* pJV53H cells were electroporated in 0.2 cm cuvettes (Bio-Rad) and applied a 2,500 V, 25 µF, 1,000 Ω pulse in a GenePulser Xcell electroporator (Bio-Rad) with a mixture of 500 ng of the oligonucleotide carrying the desired mutations and 100 ng of an oligo targeting *rpsL* that confers Sm resistance. Bacteria were then recovered in 1 mL of fresh medium, incubated overnight at 37°C, and selected on LB agar plates containing 20 µg/mL Sm. Colonies were screened for the presence of the desired point mutation by colony PCR after digestion of the PCR products with either *Xba*I or *Bsp*TI (Thermo Fisher) for 1 hour. Positive colonies, containing a mixture of bacteria carrying wild-type and mutant alleles, were streaked onto LB agar plates to isolate isogenic colonies, which were then rescreened using the same methodology.

Multiple mutations were introduced using 500 ng of pBP10 plasmid (44) for co-selection instead of the oligonucleotide conferring Sm resistance. Km-resistant colonies were screened as above described for single point mutations. Finally, once a colony with the desired mutation was obtained, pBP10 was cured by doing a passage in a medium without Km and seeding bacteria onto LB agar plates. Curation was verified using PCR with oligos targeting the Km resistance cassette.

## Analysis of *M. smegmatis* lipids and mycolic acids by TLC

Cells were grown by shaking at 37°C in 7H9 broth supplemented with 10% Albumin-Dextrose-Catalase in the presence or absence of 0.05% Tween 80. After cultivation, cells were harvested by centrifugation (2,200× RCF, 20 min). Lipids were extracted from cell pellets by extraction with 6 mL chloroform/methanol (1:2) at 56°C for 2 h, followed by two extractions in 6 mL chloroform/methanol (2:1) under the same conditions. Extracts were pooled into a glass tube, dried under a stream of nitrogen, and subjected to biphasic wash in chloroform/methanol/water (4:2:1), as described by Folch (45). After washing, organic phases from each sample were dried and dissolved in 600 µL chloroform/methanol (2:1) per 0.5 g of wet cell weight. Then, 10 µL of each sample was loaded on TLC silica gel 60 $F_{254}$ plates (Merck) and lipids were separated in following eluents— solvent I (chloroform/methanol/water 20:4:0.5), solvent II (chloroform/methanol/ammonium hydroxide/water 65:25:0.5:4), and solvent III (petroleum ether/ethyl acetate 98:2; three runs). Lipids were visualized by soaking the plates in 10% $CuSO_4$ in an 8% phosphoric acid solution and charring them with a heat gun.

For isolation of glycopeptidolipids, 50 µL from obtained lipid extracts was dried and subjected to alkaline methanolysis (200 µL 0.2 M NaOH in methanol for 2 h at 37°C) to cleave ester-linked fatty acids (46). After pH neutralization with glacial acetic acid, samples were dried under nitrogen flow and alkaline-stable GPLs were partitioned by a biphasic system composed of chloroform/methanol/water (4:2:1) (45). Organic phase was washed twice with chloroform/methanol/water (3:48:47), dried under the stream of nitrogen and resuspended in 50 µL chloroform/methanol (2:1). Then, 10 µL from purified GPLs was applied on silica gel 60 $F_{254}$ plates, separated in solvent IV (chloroform/methanol 9:1), and visualized by soaking the plates in solution of orcinol in sulfuric acid followed by charring.

Methyl esters of mycolic acids (MAME) and fatty acids (FAME) were prepared from whole cells as previously described (47). Briefly, 2 mL of 15% tetrabutylammonium hydroxide solution was added to pelleted cells and samples were saponified overnight at 100°C. After cooling down, samples were derivatized to corresponding methyl esters by 4 h incubation in the mixture of 3 mL dichloromethane, 300 µL iodomethane, and 1 mL water at room temperature on a rotary shaker. Methylated samples were washed with water three times and the resulting organic phase was dried under nitrogen flow. MAME and FAME were extracted to 3 mL diethyl ether and dried. Samples were dissolved in chloroform/methanol (2:1) and loaded on TLC plates as described for lipid extracts. Different forms of methyl esters were separated in solvent V (n-hexane/ethyl acetate 95:5; three runs) and visualized as previously stated for lipids.

## ACKNOWLEDGMENTS

We thank Katarína Mikušová from the Comenius University in Bratislava for helpful discussions during the performance of this work, and Ana Picó and Begoña Gracia from Universidad de Zaragoza for technical support. Porin-deficient M. smegmatis strains were a gift from Michael Niederweis from the University of Alabama at Birmingham.

This research was supported by internal group funding (J.A.A. and S.R.-G) and by a fellowship from the Spanish Government (Programa de Formación de Profesorado Universitario) Ref. FPU18/03873 to J.M.E.-A. J.M.E.-A. also received support for a short-term stage from the Red Temática de Excelencia de Biología de Sistemas de Micobacterias (Myconet) to I.C. J.B. received funding from Ministerio de Ciencia (MCIN/AEI/ 10.13039/501100011033) (Grant PID2020-112865RB-I00). J.K. received funding from the Slovak Research and Development Agency [grant n. APVV-19-0189] and the OPII, ACCORD, ITMS2014+: 313021X329, co-financed by ERDF.

## AUTHOR AFFILIATIONS

[1]Department of Microbiology, Pediatrics, Radiology and Public Health, Faculty of Medicine, and BIFI, University of Zaragoza, Zaragoza, Spain

[2]Department of Biochemistry, Faculty of Natural Sciences, Comenius University in Bratislava, Bratislava, Slovakia

[3]Genomics of (Re)Emerging Pathogens, Genomics and Health Area, FISABIO–Public Health, Valencia, Spain

[4]Spanish Network for Research on Epidemiology and Public Health (CIBERESP), Carlos III Health Institute, Madrid, Spain

[5]National Centre of Biotechnology (CNB-CSIC), Madrid, Spain

[6]Tuberculosis Genomics Unit, Institute for Biomedicine of Valencia (IBV-CSIC), Valencia, Spain

[7]Spanish Network for Research on Respiratory Diseases (CIBERES), Carlos III Health Institute, Madrid, Spain

[8]Research and Development Agency of Aragon Foundation (Fundación ARAID), Zaragoza, Spain

## PRESENT ADDRESS

José Manuel Ezquerra-Aznárez, Department of Microbiology, University of Washington, Seattle, Washington, USA

Henrich Gašparovič, Centre National de la Recherche Scientifique, Institut de Pharmacologie et de Biologie Structurale, Toulouse, France

## AUTHOR ORCIDs

José Manuel Ezquerra-Aznárez  http://orcid.org/0000-0001-9771-4912
Jesús Blázquez  http://orcid.org/0000-0003-0495-3848
Iñaki Comas  http://orcid.org/0000-0001-5504-9408
José A. Aínsa  http://orcid.org/0000-0003-2076-844X
Santiago Ramón-García  http://orcid.org/0000-0002-8480-0325

## FUNDING

| Funder | Grant(s) | Author(s) |
|---|---|---|
| Spanish Government (Programa de Formación de Profesorado Universitario) | FPU18/03873 | José Manuel Ezquerra-Aznárez |
| Ministerio de Ciencia e Innovación (MCIN) | PID2020-112865RB-I00 | Jesus Blazquez |
| Red Temática de Excelencia de Biología de Sistemas de Micobacterias | Myconet | Iñaki Comas |
| Slovak Research and Development Agency | APVV-19-0189 | Jana Korduláková |
| University of Zaragoza | Internal Funds | José A. Aínsa |
| | | Santiago Ramón-García |

## AUTHOR CONTRIBUTIONS

José Manuel Ezquerra-Aznárez, Conceptualization, Formal analysis, Investigation, Writing – original draft, Writing – review and editing | Henrich Gašparovič, Formal analysis, Investigation, Writing – original draft, Writing – review and editing | Álvaro Chiner-Oms, Formal analysis | Ainhoa Lucía, Resources, Supervision | Jesús Blázquez, Resources | Iñaki Comas, Supervision | Jana Korduláková, Investigation, Supervision, Writing – original draft, Writing – review and editing | José A. Aínsa, Supervision, Writing – review and editing | Santiago Ramón-García, Conceptualization, Funding acquisition, Project administration, Supervision, Writing – original draft, Writing – review and editing

## DATA AVAILABILITY

WGS data were submitted to SRA under accession number PRJNA1188729. All other relevant data are available upon request.

## ADDITIONAL FILES

The following material is available online.

### Supplemental Material

**Supplemental figures and tables (Spectrum02332-24-S0001.docx).** Fig. S1 to S3; Tables S1 to S9.

## Open Peer Review

**PEER REVIEW HISTORY (review-history.pdf).** An accounting of the reviewer comments and feedback.

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
