## [Reviewer comments · Microbiology Spectrum]

Microbiology Spectrum

Emergence of resistance to the antiparasitic selamectin in *Mycobacterium smegmatis* is improbable and contingent on cell wall integrity

José Manuel Ezquerra-Aznárez, Henrich Gašparovič, Alvaro Chiner-Oms, Ainhoa Lucía, Jesus Blazquez, Iñaki Comas, Jana Korduláková, José Aínsa, and Santiago Ramon-Garcia

Corresponding Author(s): Santiago Ramon-Garcia, Universidad de Zaragoza

Review Timeline:

Submission Date:	September 16, 2024
Editorial Decision:	October 31, 2024
Revision Received:	December 10, 2024
Accepted:	January 18, 2025

Editor: G. Marcela Rodriguez

Reviewer(s): Disclosure of reviewer identity is with reference to reviewer comments included in decision letter(s). The following individuals involved in review of your submission have agreed to reveal their identity: Mahadi Hassan Mahmoud Abdallah (Reviewer #1); Tanjore S Balganesesh (Reviewer #2)

Transaction Report:

DOI: <https://doi.org/10.1128/spectrum.02332-24>

Re: Spectrum02332-24 (Emergence of resistance to the antiparasitic selamectin in *Mycobacterium smegmatis* is improbable and contingent on cell wall integrity)

Dear Dr. Santiago Ramon-Garcia:

Thank you for the privilege of reviewing your work. Based on the reviewers' feedback and my assessment, I invite you to resubmit a revised version. Below are my comments, instructions from the Spectrum editorial office, and the reviewer's comments. Reviewer #1 proposes additional studies. Although acceptance of the manuscript is not contingent on those studies, integrating some of their suggestions in the discussion will improve that section of the manuscript.

Editor Comments

1. Please include references that support the statements in lines 95-97 about ivermectin activities.
2. Selamectin is the avermectin with the best activity against *M. ulcerans*. Is it also the most active against *M. tuberculosis*? Please expand on the criteria for selecting selamectin in the introduction; why this particular avermectin? Also, please comment on whether Selamectin is equally active against *M. tuberculosis* and *M. smegmatis*.
3. Please elaborate on the observations described in lns 104 to 107. What mycobacterium was used in the referred experiments, and what evidence supports additional targets?
4. The authors must recognize in the introduction that although providing valuable insight, their observations are restricted to *M. smegmatis* and not necessarily translatable to *M. tuberculosis*.
5. According to Table 1. The addition of tyloxapol in the medium increases the MIC 16-fold. The authors don't interpret or follow up on this result. Are there reasons to think the detergent inactivates the antibiotic? Or modifying the bacterial cell envelope by the detergent make the bacteria more tolerant to selamectin? This result can be related to their findings about cell envelope integrity in the discussion.
6. When first mentioned, please define the functionality of genes mentioned, such as *mps1*, *mmpL11*, *mshA*, *embA* etc.
7. Define what FICI stands for in the main text.
8. The Methods' text should include the Characteristics and source of the plasmids used.
9. The figure legends must include the number of biological replicates and statistical significance.

Revision Guidelines

- Upload point-by-point responses to the issues raised by the editor and reviewers in a file named "Response to Reviewers," NOT in your cover letter.
- Upload a compare copy of the manuscript (without figures) as a "Marked-Up Manuscript" file.
- Upload a clean .DOC/.DOCX version of the revised manuscript and remove the previous version.
- Each figure must be uploaded as a separate, editable, high-resolution file (TIFF or EPS preferred), and any multipanel figures must be assembled into one file.
- Any supplemental material intended for posting by ASM should be uploaded with their legends separate from the main manuscript. You can combine all supplemental material into one file (preferred) or split it into a maximum of 10 files with all associated legends included.

Data availability: ASM policy requires that data be available to the public upon online posting of the article, so please verify all

links to sequence records, if present, and make sure that each number retrieves the full record of the data. If a new accession number is not linked or a link is broken, provide Spectrum production staff with the correct URL for the record. If the accession numbers for new data are not publicly accessible before the expected online posting of the article, publication may be delayed; please contact production staff (Spectrum@asmusa.org) immediately with the expected release date.

Sincerely,
G. Marcela Rodriguez
Editor
Microbiology Spectrum

Reviewer #1 (Comments for the Author):

Further validation and insights on the findings would add depth to the study and are quite relevant. Here's how they can be applied and integrated with the current understanding:

In Vivo Testing: Testing selamectin's efficacy on resistant strains in animal models would be valuable for translating the study's findings into clinical relevance. By comparing outcomes between resistant and non-resistant strains, one could validate if the observed inactivation impacts selamectin's efficacy in complex biological systems. This step would address how mutations or resistance mechanisms might play out in real infections.

Selamectin Binding Assay: Conducting binding assays like affinity chromatography or surface plasmon resonance (SPR) could reveal whether selamectin's binding affinity to specific mycobacterial cell targets differs in resistant strains. Reduced binding could indicate that mutations or structural changes in resistant strains prevent effective drug action, suggesting a direct mechanism of inactivation.

Role of Mycobacterial Envelope Integrity: The study's discovery that an intact envelope is essential for resistance mechanisms highlights an area for further research. Investigating how envelope disruptions make mycobacteria more vulnerable to selamectin could help elucidate the role of envelope integrity in antimicrobial susceptibility across other mycobacteria as well.

DprE1 Inhibitors for Contextual Comparison: Since the DprE1 enzyme is essential for cell wall synthesis in mycobacteria, inhibitors targeting this enzyme offer a useful comparison. With ~1519 DprE1 inhibitors identified and characterized from 2009 to April 2022, a comparison of their physicochemical properties and ADMET profiles with selamectin could reveal patterns in efficacy, resistance potential, and toxicity. Such comparisons could clarify why selamectin's mechanism might require envelope integrity, unlike other mycobacterial inhibitors.

Reviewer #2 (Comments for the Author):

The paper elegantly describes the construction of several genetic constructs of *M. smegmatis* and its interaction with the anti-parasitic compound Selamectin. Mutants were selected on a *M. smeg.* strain which carries a 'mutator' phenotype, mutants were mapped to *mps* and *mml* genes, one involved in *gpl* and the other on the transporter. The interactions of these mutants with Ethambutol is also interesting.

The relevance of this work for *Mtb* in terms of understanding the selamectin MOA is not there.

The following points are unclear and need to be defined

1. The paper discusses several cell wall antibiotics that may have an impact, if their hypothesis is that these cell wall antibiotics do have an effect with selamectin as they all impact the cell wall then further substantiation is required.
2. The restoration of selamectin sensitivity in *EmbR* mutants is rather counterintuitive- a intact cell wall should have more of an effect in preventing uptake, the opposite is true and the authors do not clarify. Perhaps the '*gpl*' profile of these 'sensitive' mutants could throw some light. Other interactions that explain this phenotype need to be studied.
3. Do the low level selamectin resistance mutants also show synergy with the cell-wall inhibitors.

Review: Emergence of Resistance to the Antiparasitic Selamectin in *Mycobacterium smegmatis* Is Improbable and Contingent on Cell Wall Integrity

Introduction

This review examines the study's insights into selamectin, a member of the avermectin family, and its antimycobacterial action using *Mycobacterium smegmatis* as a surrogate for *M. tuberculosis*. The study presents a novel understanding of selamectin's efficacy and highlights the role of the mycobacterial envelope in resistance mechanisms. Here, I propose several validation steps and further analyses that could strengthen the study's findings and offer additional clarity on selamectin's mechanism of action.

Key Suggestions for Additional Experiments

1. In Vivo Testing

Testing selamectin's efficacy on resistant strains in animal models would be valuable to assess its clinical relevance. By comparing outcomes between resistant and non-resistant strains, we could determine if the observed inactivation significantly impacts selamectin's effectiveness in complex biological systems. This would simulate real infection dynamics, clarifying how resistance mutations affect selamectin efficacy in vivo.

2. Selamectin Binding Assay

Conducting binding assays, such as affinity chromatography or surface plasmon resonance (SPR), would help determine if selamectin's binding affinity to mycobacterial targets differs in resistant strains. Reduced binding in resistant strains could indicate mutations or structural changes that reduce selamectin efficacy, suggesting a direct mechanism of drug inactivation. This would provide a molecular basis for understanding how resistance mechanisms alter selamectin action.

3. Role of Mycobacterial Envelope Integrity

The study's finding that an intact mycobacterial envelope is crucial for selamectin susceptibility points to a unique area for further research. Investigating how envelope disruptions affect mycobacterial vulnerability to selamectin could yield insights into the role of envelope integrity in antimicrobial susceptibility across different mycobacterial species.

Comparative Analysis with DprE1 Inhibitors

DprE1 inhibitors provide an ideal comparison, as the DprE1 enzyme is essential for mycobacterial cell wall synthesis. With 1519 DprE1 inhibitors characterized from 2009 to April 2022, an analysis of their physicochemical properties and ADMET profiles against selamectin could identify patterns in efficacy, resistance, and toxicity. This comparison may elucidate why selamectin's mechanism requires envelope integrity, unlike some other mycobacterial inhibitors, and inform future drug design strategies.

Conclusion

In summary, these additional analyses could enhance our understanding of selamectin's antimycobacterial mechanism, confirming its reliance on envelope integrity and potentially guiding new strategies for drug development. Expanding on these findings can strengthen the study's impact, contributing to our knowledge of antimicrobial resistance and the development of targeted therapies for mycobacterial infections.

Dr. Mahadi Abdallah

Associate Professor of Microbiology

Dean MLS- Imperials college

Answers to editor's comments - Spectrum02332-24

Emergence of resistance to the antiparasitic selamectin in Mycobacterium smegmatis is improbable and contingent on cell wall integrity

Editor's comments

1. Please include references that support the statements in lines 95-97 about ivermectin activities.

Answer (A) 1. Reference 15 supports the statement about new potential uses for ivermectin.

2. Selamectin is the avermectin with the best activity against *M. ulcerans*. Is it also the most active against *M. tuberculosis*? Please expand on the criteria for selecting selamectin in the introduction; why this particular avermectin? Also, please comment on whether Selamectin is equally active against *M. tuberculosis* and *M. smegmatis*.

A2. Selamectin is the most active avermectin against most mycobacterial strains, including *M. tuberculosis*. This is supported by references [9-11] in the manuscript. In addition to its *in vitro* potency, we selected selamectin as the model avermectin because it displays better toxicology profile compared to other avermectins [11]. We have included this in lines 111-118 of the latest version of the manuscript.

Selamectin has comparable activity in *M. tuberculosis* and *M. smegmatis*. MIC values for these two strains were first reported in reference [9]. We have added this statement in lines 123-124.

3. Please elaborate on the observations described in lns 104 to 107. What mycobacterium was used in the referred experiments, and what evidence supports additional targets?

A3. Enzymatic assays were conducted using the purified *M. tuberculosis* DprE1 enzyme. Whole cell assays were carried out in *M. smegmatis*. We also performed lipid analysis of *M. smegmatis* and *M. tuberculosis* cells treated with selamectin, which did not show any changes (e.g., accumulation of trehalose monomycolates and trehalose dimycolates) indicating DprE1 inhibition.

We have modified lines 104-107 to describe the findings from reference [18] in more detail.

4. The authors must recognize in the introduction that although providing valuable insight, their observations are restricted to *M. smegmatis* and not necessarily translatable to *M. tuberculosis*.

A4. We have modified the final paragraph of the introduction (lines. 103-110) to acknowledge this.

5. According to Table 1. The addition of tyloxapol in the medium increases the MIC 16-fold. The authors don't interpret or follow up on this result. Are there reasons to think the detergent inactivates the antibiotic? Or modifying the bacterial cell envelope by the detergent make the bacteria more tolerant to selamectin? This result can be related to their findings about cell envelope integrity in the discussion.

A5. We appreciate this suggestion. We have several hypotheses that could explain why tyloxapol interferes with selamectin activity. First, selamectin is sparingly soluble in aqueous media, and tyloxapol could form complexes with selamectin, resulting in a lower effective concentration of selamectin in the medium. It is also possible that tyloxapol is directly affecting the lipid composition of the outer mycobacterial membrane, and this impacts the susceptibility to selamectin. This second hypothesis would be consistent with the intermediate resistance phenotype being driven by changes in the lipid composition of the outer membrane.

6. When first mentioned, please define the functionality of genes mentioned, such as mps1, mmpL11, mspA, and mshA, embA etc.

A6. We have introduced a brief description of each gene function when first mentioned.

7. Define what FICI stands for in the main text.

A7. This has now been corrected.

8. The Methods' text should include the Characteristics and source of the plasmids used.

A8. We have created a new table (table S8) in the supporting information section summarizing the plasmids used in this study. We have also added the references corresponding to the source of the plasmids in the main text.

9. The figure legends must include the number of biological replicates and statistical significance.

A9. This has now been corrected.

Answers to reviewer's comments

- **Reviewer 1.**

Further validation and insights on the findings would add depth to the study and are quite relevant.

Here's how they can be applied and integrated with the current understanding:

In Vivo Testing: Testing selamectin's efficacy on resistant strains in animal models would be valuable for translating the study's findings into clinical

relevance. By comparing outcomes between resistant and non-resistant strains, one could validate if the observed inactivation impacts selamectin's efficacy in complex biological systems. This step would address how mutations or resistance mechanisms might play out in real infections.

Selamectin Binding Assay: Conducting binding assays like affinity chromatography or surface plasmon resonance (SPR) could reveal whether selamectin's binding affinity to specific mycobacterial cell targets differs in resistant strains. Reduced binding could indicate that mutations or structural changes in resistant strains prevent effective drug action, suggesting a direct mechanism of inactivation.

Role of Mycobacterial Envelope Integrity: The study's discovery that an intact envelope is essential for resistance mechanisms highlights an area for further research. Investigating how envelope disruptions make mycobacteria more vulnerable to selamectin could help elucidate the role of envelope integrity in antimicrobial susceptibility across other mycobacteria as well.

DprE1 Inhibitors for Contextual Comparison: Since the DprE1 enzyme is essential for cell wall synthesis in mycobacteria, inhibitors targeting this enzyme offer a useful comparison. With ~1519 DprE1 inhibitors identified and characterized from 2009 to April 2022, a comparison of their physicochemical properties and ADMET profiles with selamectin could reveal patterns in efficacy, resistance potential, and toxicity. Such comparisons could clarify why selamectin's mechanism might require envelope integrity, unlike other mycobacterial inhibitors.

A10. We appreciate Reviewer 1's suggestions. We are currently investigating the mechanism of action of avermectins in *Mycobacterium tuberculosis* and the potential of selamectin for TB therapy. All the above remarks are very insightful and will incorporate these recommendations into our current and future experimental plans.

- **Reviewer 2.**

The paper elegantly describes the construction of several genetic constructs of *M. smegmatis* and its interaction with the anti-parasitic compound Selamectin. Mutants were selected on a *M. smeg.* strain which carries a 'mutator' phenotype, mutants were mapped to *mps* and *mml* genes, one involved in *gpl* and the other on the transporter. The interactions of these mutants with Ethambutol is also interesting. The relevance of this work for *Mtb* in terms of understanding the selamectin MOA is not there.

The following points are unclear and needs to be defined

1. The paper discusses several cell wall antibiotics that may have an impact, if their hypothesis is that these cell wall antibiotics do have an effect with selamectin as they all impact the cell wall then further substantiation is required.

A11. We appreciate Reviewer's 2 comments. We used cell wall inhibitors that target different layers of the mycolyl-arabinogalactan-peptidoglycan complex. If the interaction between selamectin and these inhibitors were exclusively due to an increased permeability across the cell wall, we should expect similar changes in the MIC in both the wild-type and mutant backgrounds. However, our findings do not match this hypothesis. Instead, we observed that

ethambutol, vancomycin and cefradine had a differential effect in the SEL-R background. This suggests that the effect of these compounds goes beyond increasing selamectin exposure and that the integrity of this structure is required to sustain the mechanism that confers high-level selamectin resistance.

2. The restoration of selamectin sensitivity in EmbR mutants is rather counterintuitive- a intact cell wall should have more of an effect in preventing uptake, the opposite is true and the authors do not clarify. Perhaps the 'gpl' profile of these 'sensitive' mutants could throw some light. Other interactions that explain this phenotype needs to be studied.

A12. We did not evaluate the cell wall in the ethambutol-resistant mutants generated in this study and, therefore, cannot compare the integrity to the wild-type *M. smegmatis*. Given that the *embB* mutations identified in the EMB-R mutants with restored susceptibility to selamectin have not been reported in *M. tuberculosis* clinical isolates, we speculate that these *embB* mutations could weaken the cell wall and increase the susceptibility to other stresses, including selamectin exposure. On the other hand, we expect that the *embB* mutation described in *M. tuberculosis* EMB-R clinical isolates does not have such an impact on overall fitness. We believe this is consistent with previous observations in which mutations that confer resistance but have significant fitness costs are not selected *in vivo*, such as *katG* nonsense mutations in the presence of isoniazid.

3. Do the low level selamectin resistance mutants also show synergy with the cell-wall inhibitors.

A13. We have not tested the SEL-I parental strains for interactions with cell wall inhibitors other than ethambutol, for which we observed a behavior similar to the wild-type *M. smegmatis* strain. We believe subinhibitory concentrations of some cell wall inhibitors can restore susceptibility to the basal level by interfering directly with the mechanism of resistance but they cannot increase the susceptibility—at least in assays that use MIC as the readout—beyond the basal level.

Re: Spectrum02332-24R1 (Emergence of resistance to the antiparasitic selamectin in *Mycobacterium smegmatis* is improbable and contingent on cell wall integrity)

Dear Dr. Santiago Ramon-Garcia:

Your manuscript has been accepted, and I am forwarding it to the ASM production staff for publication. Your paper will first be checked to make sure all elements meet the technical requirements. ASM staff will contact you if anything needs to be revised before copyediting and production can begin. Otherwise, you will be notified when your proofs are ready to be viewed.

Sincerely,
G. Marcela Rodriguez
Editor
Microbiology Spectrum